# Niche Partitioning with Temperature among Heterocystous Cyanobacteria (*Scytonema* spp., *Nostoc* spp., and *Tolypothrix* spp.) from Biological Soil Crusts

**DOI:** 10.3390/microorganisms8030396

**Published:** 2020-03-12

**Authors:** Ana Giraldo-Silva, Vanessa M. C. Fernandes, Julie Bethany, Ferran Garcia-Pichel

**Affiliations:** 1School of Life Sciences, Arizona State University, Tempe, AZ 85287, USA; amgirald@asu.edu (A.G.-S.); jabethan@asu.edu (J.B.); 2Center for Fundamental and Applied Microbiomics (CFAM), Biodesing Institute, Arizona State University, Tempe, AZ 85287, USA

**Keywords:** biological soil crust, drylands, niche partitioning, nitrogen fixing cyanobacteria

## Abstract

Heterocystous cyanobacteria of biocrusts are key players for biological fixation in drylands, where nitrogen is only second to water as a limiting resource. We studied the niche partitioning among the three most common biocrust heterocystous cyanobacteria sts using enrichment cultivation and the determination of growth responses to temperature in 30 representative isolates. Isolates of *Scytonema* spp. were most thermotolerant, typically growing up to 40 °C, whereas only those of *Tolypothrix* spp. grew at 4 °C. *Nostoc* spp. strains responded well at intermediate temperatures. We could trace the heat sensitivity in *Nostoc* spp. and *Tolypothrix* spp. to N_2_-fixation itself, because the upper temperature for growth increased under nitrogen replete conditions. This may involve an inability to develop heterocysts (specialized N_2_-fixing cells) at high temperatures. We then used a meta-analysis of biocrust molecular surveys spanning four continents to test the relevance of this apparent niche partitioning in nature. Indeed, the geographic distribution of the three types was clearly constrained by the mean local temperature, particularly during the growth season. This allows us to predict a potential shift in dominance in many locales as a result of global warming, to the benefit of *Scytonema* spp. populations.

## 1. Introduction

In drylands, where plant growth is limited by water and nutrients, the soil surface can be occupied by communities of microorganisms known as biological soil crusts (biocrusts; see [1] for a primer, and [2] for a monograph), which play crucial roles for the fertility and stability of drylands. Their presence enhances resistance to erosion caused by water [3] or wind [4,5], modifies soil surface temperature [6], and influences water retention and runoff [7,8,9]. Colonization of bare soils, typically pioneered by highly motile filamentous cyanobacteria like *Microcoleus vaginatus* and *Microcoleus steenstrupii* [10] results in the formation of incipient communities. Once the surface is stabilized, sessile, heterocystous cyanobacteria colonize secondarily. The community also hosts a variety of populations of heterotrophic bacteria [11,12], archaea [13], and fungi [14] as well as lichens and mosses [15], which are typical of the most developed crusts. Once established, these heterocystous cyanobacteria are significant contributors to dinitrogen inputs in soils crusts [16], taking over this role from the heterotrophic diazotrophic bacteria [17] that enter in C for N symbioses with *M. vaginatus* in early succession stages [18]. Three phylogenetically well-defined clades, *Scytonema* spp., *Nostoc* spp. and *Tolypothrix/Spirirestis* spp., have been identified as the most abundant diazotrophic cyanobacteria in biocrusts communities of the Southwestern US [19]. Soil crusts are typically in a perennial state of N deficiency because the internal N cycle is broken (denitrification is apparently absent from most biocrusts; [20,21]). Biological fixation thus remains a necessity for continued growth. Fixed atmospheric C and N [20,22,23], along with other elements [24] can then be exported to underlying soils, improving landscape soil fertility. Because drylands cover nearly 45% of the total Earth continental area [25], and aridity is predicted to increase due to global warming [26,27,28], this N export activity of biocrusts matters not only locally, but also globally. In fact, the global N_2_-fixation of cryptogamic covers, much of which are biocrusts, has been estimated at 49 Tg/yr, accounting for nearly 50% of the biological N_2_-fixation on land [29].

N_2_-fixation activity has been determined experimentally to be optimal in the range of 15–30 °C regardless of the biocrusts origin or successional stage assayed in the US Southwest [30,31], with rates decreasing significantly between 30 and 35 °C [31]. This sensitivity has been ascribed to possible deleterious effects of temperature on N_2_-fixing cyanobacteria [31]. Thermophysiological studies using laboratory isolates [31,32,33] or geographical distribution in molecular tallies [34] have shown that the three main clades of biocrust heterocystous cyanobacteria are characterized by different temperature ranges for growth: the *Scytonema* spp. clade tends to be more thermotolerant, whereas the *Tolypothrix* spp. clade shows psychrophilic preferences, and strains in the *Nostoc* spp. clade shows a preference for mild temperatures (15 to 30 °C). However, these results come from the evaluation of a restricted number of sites or strains, and the patterns were not always robust. Clearly, however, the results point to a potential for differential sensitivity of these cyanobacteria to environmental warming, a future scenario with which biocrust will have to contend. Drylands at large will likely become warmer and drier in response to global warming. In particular, the US Southwest is predicted to experience an increase in temperature of about 1 °C per decade [26], accompanied by alterations in precipitation frequency [35,36,37].

In this contribution we wanted to evaluate in detail the thermophysiology of biocrust heterocystous cyanobacteria using cultivated isolates, and to test if their niche differentiation is regulated by N_2_-fixation. Finally, we wanted to test if the physiological data obtained from cultures, can explain the current biogeographic distribution of each clade, and hence potentially help us predict their fate in the face of global warming. Our results show that these cyanobacteria show markedly different thermophysiological patterns in culture and consistent world-wide distributions in nature. This points to a potential for differential sensitivity among them to global warming, allowing us to predict a microbial replacement that biocrusts will have to contend with in future climate change scenarios.

## 2. Materials and Methods

### 2.1. Enrichment Cultures

Field biocrusts were collected from the (cold) Great Basin Desert (Utah, USA), and from the (warm) Chihuahuan desert (New Mexico, USA), and from two soil textural types in each, Great Basin: sandy clay loam and clay loam, and Chihuahuan: clay loam and loamy sand. Locations and soil types details are given in [38]. Three enrichment cultures were prepared from each site and incubation temperature by randomly placing small biocrust crumbles and spreading it over 1.5 % (*w*/*v*) agar-solidified minimal medium without combined nitrogen (BG11_0_; [39,40]) in Petri dishes. They were incubated at 4, 25 and 30 °C, for 20 days, under 20 to 27 μmoL m^−2^ s^−1^ light from fluorescent bulbs under a 14 h photoperiod. After incubation, colonies were counted, sampled and observed under the compound microscope (labophot-2, Nikon, Tokyo, Japan) to be assigned to one of the three morphotypes. Differences in the relative proportions were assessed via permutational multivariate analysis of variance (PERMANOVA). PERMANOVAS were performed on the Bray-Curtis distance matrices of relative proportions derived from colonies counts and used 999 permutations. PERMANOVAS were run on PRIMER 6 software with PERMANOVA+ add on [41,42].

### 2.2. Experimental Organisms and Growth Conditions

Thirty cyanobacterial strains: 12 *Scytonema* spp., 10 *Nostoc* spp., and eight *Tolypothrix* spp. previously isolated as a part of our “microbial biocrust nurseries” protocols (see [38] as well as a description of the cyanobacterial community structure of the biocrust communities of origin), were used in our experiments. Briefly, strains were isolated from enrichment cultures in agar-solidified BG11_0_ Petri plates followed by multiple streaking of colonies on fresh agar plates. Strain identity was first assessed by microscopy, and then confirmed by PCR amplification of the V4 region of the 16S rRNA gene using cyanobacteria specific primers CYA359F/CYA781R [43] (PCR protocol therein), blast comparisons, and by placing the sequences on the cyanobacterial tree Cydrasil (https://itol.embl.de/tree/1491698589270801574806192). PCR products were sequenced using Sanger sequencing. All strains were unicyanobacterial, are kept in our local culture collection, and are available upon request. Strain accession numbers along with their denomination coding for site of origin can be found in Appendix A. Stock cultures were grown in 175 mL cell culture flasks containing 100 mL of medium free of combined nitrogen (BG11_0_). Cultures were maintained at 25 ± 2 °C, under a 14 h photoperiod, illuminated at 20–27 μmoL (photon) m^−2^ s^−1^ provided by white fluorescent tubes.

### 2.3. Delineation of Temperature Range for Growth and Survival of Isolates

Prior to inoculation, stock liquid cultures of each strain were homogenized by repeatedly forcing biomass through a 60 mL sterile syringe, and immediately washed with fresh BG11_0_ medium by five consecutive centrifugations (8 min, 8437 g, 25 °C). Aliquots of this homogenized cultures served as inoculum (5% *v*/*v*) for experimental cultures, which were run on 50 mL cell culture flasks filled to the 20 mL mark. Each strain was incubated at 4, 15, 25, 30, 35, 40 and 45 °C in triplicate, exposed to a light intensity of 20–27 μmoL (photon) m^−2^ s^−1^ provided by white fluorescent tubes, in a 12 h photoperiod regime. Growth was estimated visually after 30 days as either positive for growth (there was an obvious increase in biomass at the end of the incubation period compared to initial inoculum) or negative for growth (either no-growth (stasis) or patent death). Assays assigned to “no-growth” looked healthy, with brightly pigmented cells, but did not show appreciable biomass increase during the incubation, whereas assays assigned to patent death exhibited a total loss of pigmentation. The whole experiment was replicated a second time in full, and growth in any of the trials was reported as positive.

### 2.4. Influence of Diazotrophy on the Upper Temperature Limit for Growth

A homogenized, cleaned culture mix was prepared for each of the strains as detailed above, and inoculated (5% *v*/*v*) in 50 mL cell culture flasks containing either medium without combined nitrogen (BG110) or nitrogen-containing medium (BG11). Triplicate cultures were incubated at 35 and 40 °C, illuminated with 20–27 μmol (photon) m^−2^ s^−1^ provided by white fluorescent tubes, in a 12 h photoperiod regimen, for 30 days.

### 2.5. Heterocyst and Vegetative Cell Counts

To determine the frequency of heterocysts we conducted microscopic cell counts on fresh wet mounts under bright field illumination in a Nikon labophot-2 compound microscope. At least 200 cells were counted in each determination. To determine the effect of nitrogen source and incubation temperature on heterocyst frequency we examined triplicate cultures of each strain at 25, 35 and 40 °C, all at day 7 after inoculation, time at which all tested strains appeared healthy. The full experiment was replicated for a total *n* = 6.

### 2.6. Chlorophyll a Determination

Chlorophyll *a* (Chl *a*) was measured as a proxy for phototrophic biomass. Chl *a* was extracted in triplicate, in 90% acetone, according to [44], vortexed for 30 s. and allowed to extract for 24 h at 4 °C in the dark. Extracts were clarified by centrifugation (5 m at 8437 g). Absorbance spectra of the clarified extracts was recorded on a UV-visible spectrophotometer (UV-1601, Shimadzu, Kyoto, Japan). Interference from scytonemin and carotenoids was corrected using the trichromatic equation of [45].

### 2.7. Meta-analysis of Temperature Niches

In an attempt to look for a temperature segregation pattern among the studied taxa in the natural biocrust environment, we performed a meta-analysis of all bacterial 16S rRNA tallies available publicly. We performed a literature search, and either downloaded from public databases or directly requested raw sequence data from authors from multiple environmental biocrust surveys conducted at different locations around the world. We collected data from different arid and semiarid regions in USA [6,34,46,47,48], Mexico [33] and Australia [49], from arid, semiarid and alpine regions in Europe [32,50], from the arid Gurbantunggut desert in China [51], and from the Brazilian savannah (Cerrado) [52]. A complete list of the biocrust surveys with locations, environmental variables, and other relevant information can be found in Appendix A.

For all but the dataset from [34], forward reads obtained with pyrosequencing [51] and paired-end reads obtained with Illumina were demultiplexed, and quality controlled using the DADA2 plugin [53] available in Qiime 2018.6 [54], creating a feature table containing representative sequences (features) and their frequency of occurrence. Highly variable positions were removed using MAFFT [55], and phylogenetic trees were generated using FastTree [56]. Preliminary taxonomic assignment was done using the Naïve Bayes classifier [57] trained on the Greengenes 13.8 release database [58]. For the [34] dataset, because quality files (.fastq) were not available, and in an effort to control for sequence quality before preforming any downstream analysis, raw sequences were first filtered using USEARCH 7 [59] to remove all sequences with less than 210 bp. Overall this step filtered out up to 5% of the total sequences in some but not all samples. Additionally, the first and last 10 bp of each sequence were trimmed using Fastx (http://hannonlab.cshl.edu/fastx_toolkit/). Quality controlled sequences were assigned to individual samples and barcodes were removed using Qiime 1.8 [54] using the *multiple*_*split_librairies_fastq.py* script. Operational taxonomic units (OTUs) were defined with a threshold of 97% similarity and clustered using UCLUST [59] using the *pick_open_reference_otus.py* script in Qiime. Potential chimeras, and singleton OTUs were removed from further consideration. Preliminary taxonomic assignments were done with the RDP (Ribosomal Database Project) classifier [60], and representative sequences were then aligned against the Greengenes database core reference alignment [58].

Cyanobacterial sequences (features) and OTUs were filtered out from the master file, and a more refined taxonomic assignment at the genus and species level was further informed throughout phylogenetic placements. Query cyanobacterial sequences (and OTUs) were phylogenetically placed in our cyanobacteria reference tree CYDRASIL version-0.22a (https://github.com/FGPLab/ cydrasil/tree/0.22a, accessed in July, 2019), by aligning sequences to the cyanobacterial tree alignment using PaPaRa [61], and then placing them into the reference tree using the RaxML8 Evolutionary Placement Algorithm [62]. The resulting trees were imported and visualized in the iTOL4 server [63]. Accession numbers of representative strains of the clades in which *Scytonema* spp., *Nostoc* spp. and *Tolypothrix* spp. were assigned according to CYDRASIL are included in Appendix A.

The proportion of *Scytonema* spp., *Nostoc* spp. and *Tolypothrix* spp. within the heterocystous cyanobacterial community was calculated by dividing the number of reads of either *Scytonema* spp., *Nostoc* spp. or *Tolypothrix* spp., by the sum of the number of reads of all N_2_-fixing cyanobacteria found at each location. Resulting proportions were plotted against the mean annual temperature (MAT) and the mean temperature of the wettest quarter of the year (growth season) in each location of origin. A total of 25 (out of 109) locations at which the total relative abundance of N_2_-fixing cyanobacteria was lower than 0.5 % of all reads were excluded from final plots. Mean annual temperature and mean temperature of the wettest quarter of the year were calculated from environmental variables of monthly climate data for minimum, mean, and maximum temperature and for precipitation for 1970-2000. Data was downloaded from WorldClim -Global Climate Data -version 2 (http://www.worldclim.org; [64]. Linear regressions between the proportion of sequence reads (arcsine transformed) of each taxon among heterocystous cyanobacteria and climatic parameters (MAT and MTempWetQ) were used to test significance of environmental patterns.

## 3. Results

### 3.1. Encrichment Cultivation

Enrichment cultures for diazotrophic photoautotrophs carried out at different temperatures using inoculum from four different biocrusts were very revealing. Only heterocystous cyanobacteria were enriched for in our medium free of nitrate and ammonium, and all 994 colonies examined belonged to one of the three major clades known from biocrusts: *Nostoc* spp., *Tolypothrix* spp., and *Scytonema* spp. [19], as determined by microscopic inspection. The relative proportions obtained, however, were strongly dependent on the temperature of incubation (Figure 1). The composition of the enrichments at 4 °C was significantly different from those growing at 25 °C (PERMANOVA pseudo-F: 6.22 df: 22 *p* ≥ 0.001) and 30 °C (PERMANOVA pseudo-F: 9.36 df: 22 *p* ≥ 0.001); the same was true for the comparison of 25 and 30 °C (PERMANOVA pseudo-F: 6.43 df: 22 *p* ≥0.001). *Scytonema* spp. made up the majority of the colonies at 30 °C, whereas *Tolypothrix* spp. was preferentially selected for at 4 °C. *Nostoc* spp. had a slight advantage at lower temperatures as well. This was so regardless of the origin of the crusts used for inoculation, in that there was no significant effect on outcomes by location (PERMANOVA, *p* ≤ 0.2; full dataset presented in Appendix A).

### 3.2. Temperature Range for Growth (or Survival) of Isolated Strains

All cyanobacterial strains (tested in medium without combined N) showed robust growth at 15 and 25 °C, while none grew at 45 °C (Figure 2), the lower limit of moderate thermophilly. Formally then, all these strains were mesophiles with respect to temperature. At 4 °C, all *Tolypothrix* spp. strains grew well, while only one *Scytonema* sp. strain did. At this temperature, three *Nostoc* spp. strains did not grow, while five strains were in apparent stasis (they neither grew nor show signs of cellular degradation). At 30 °C four out of eight *Tolypothrix* spp., nine out of ten *Nostoc* spp., and eleven out of twelve *Scytonema* spp. strains grew well. At 35 and 40 °C, no *Nostoc* spp. or *Tolypothrix* spp. strains grew, while eleven out of twelve *Scytonema* spp. did.

### 3.3. Upper Temperature Limit for Growth and N_2_-Fixation

We looked at growth (and survival) responses more in detail as a function of nitrogen source (N_2_-fixing vs. non N_2_-fixing conditions) in the upper range of temperature (35 and 40 °C) in an effort to infer if N_2_-fixation was the most sensitive cellular process determining the observed outcomes. Figure 3 shows the biomass yield of the 30 cyanobacterial strains after 30 days of growth cultivated in medium without combined nitrogen (nitrogen-free) and nitrogen replete media.

Our results show that providing a source of fixed nitrogen expanded the range for growth in many of them to 35 °C (*Scytonema* spp. JS003; *Nostoc* spp. HS002, HS094, HS013, *Tolypothrix* spp., HSN032, HSN033, HSN034) and in some cases, strains survived at 40 °C (*Nostoc* spp. HSN008, HS020, HS002, HS096, FB23, FB26; *Tolypothrix* sp. HSN042). The last column in Figure 3. shows the biomass yield in nitrogen replete minus that attained in medium without combined nitrogen at 35 °C, indicating a generalized positive effect on growth under nitrogen-replete conditions. For sixteen out of thirty strains this difference in growth was significant. This gives support to the contention that the upper temperature for growth may be determined by the sensitivity of N_2_-fixation in *Nostoc* spp. and *Tolypothrix* spp.; whereas it is not nearly as determinant for *Scytonema* spp.

### 3.4. Heterocyst Frequency

To determine if this effect on N_2_-fixation was perhaps due to an inability to develop heterocysts (a developmentally specialized cell type dedicated to this process), we conducted microscopic counts of vegetative cells and heterocysts in strains incubated for seven days at different temperatures (Table 1). Counts were performed only on apparently healthy filaments, but at 35 and 40 °C, biomass from all replicates of *Nostoc* sp. HSN008 and *Tolypothrix* sp. HSN042 looked yellowish, and microscopy revealed high cell mortality as well. In fact, in one occasion, one set of replicates of *Nostoc* sp. HSN008 did not survive to day 7 (Table 1). All strains grown at 25 °C looked healthy when counts were performed. Those caveats aside, the frequency of heterocysts declined precipitously for *Nostoc* spp. strains above 35 degrees, and above 30 degrees for *Tolypothrix* spp. strains. In *Scytonema* spp., there were only slight decreases in this frequency in the temperature range tested. This is consistent with a cell developmental basis for the sensitivity of N_2_-fixation to high temperatures in *Nostoc* spp. and *Tolypothrix* spp.

### 3.5. Thermal Niche of Biocrust Heterocystous Cyanobacteria through Meta-Analyses of Molecular Surveys

A total of 84 locations from eleven different biocrust surveys conducted in different arid and semiarid regions in North and South America, Europe, Australia, and China, and in the Brazil Savannah (see Appendix A), were used in a meta-analysis to assess the relative contribution of the three main clades of heterocystous cyanobacteria along temperature related parameters. Figure 4 shows the relative proportion of *Scytonema* spp., *Nostoc* spp. and *Tolypothrix* spp., plotted against the mean annual temperature (MAT) of origin and the corresponding mean temperature during the wettest quarter of the year (MTempWetQ). MTempWetQ was used as a proxy for growth season since biocrust organisms are metabolically active only when water is available [65] and are relatively insensitive to heat stress when dry. The relative abundance of *Scytonema* spp. was positively correlated with MAT (*p* = 4 × 10^−4^) and with MTempWetQ (*p* = 10^−8^); *Nostoc* spp. was negatively correlated with MAT (*p* = 5 × 10^−3^) and with MTempWetQ (*p*= 10^−7^). The relative abundance of *Tolypothrix* spp. was also negatively correlated with MAT (*p*= 0.035) and with MTempWetQ (*p* = 3 × 10^−3^). Using linear regression on arcsine transformed data, MAT explained 15, 9 and 1 of the variability in *Scytonema* spp., *Nostoc* spp. and *Tolypothrix* spp., respectively. The explanatory power of MTempWetQ was much higher in all cases, rising to 32, 28 and 11%, respectively. Based on MTempWetQ, *Scytonema* spp. could attain dominance at warmer temperatures (Figure 4A), while at lower temperatures, *Tolypothrix* spp. (Figure 4C), followed by *Nostoc* spp. (Figure 4B) attain higher maximal relative abundances. Detailed statistics are in Appendix A).

## 4. Discussion

The cyanobacteria *Scytonema* spp., *Nostoc* spp. and *Tolypothrix* spp. are secondary colonizers in the ecological succession of biocrust communities [6], where they are among the most common heterocystous organisms [6,19,32,38,50], and contribute much of the nitrogen inputs to the community at this stage of development [66]. Therefore, it is logical to assume that their presence and relative abundance have direct effects on the N_2_-fixation capability of late successional biocrusts. Using quantitative enrichment cultures we could clearly demonstrate differential fitness in these cyanobacteria at different temperatures, in a pattern that confirms the preferences inferred in prior field [32,34] and cyanobacterial cultures thermophysiological assays [32,33,34], where the biocrust cyanobacteria *M. steenstrupii* complex and *Scytonema* spp. were found to be more thermotolerant than *M. vaginatus*, and *Tolypothrix* spp. and *Nostoc* spp., respectively.

Using a set of cultivated strains (12 *Scytonema* spp., 10 *Nostoc* spp. and eight *Tolypothrix* spp.) isolated from cold and hot desert locations of the Southwestern US, the temperature range for growth revealed a pattern of niche differentiation according to temperature: *Tolypothrix* spp. strains having an advantage at the lower temperatures, and *Scytonema* spp. strains at higher temperatures. *Nostoc* spp. strains occupied only the mesic part of the temperature range. This niche separation is similar to that found in non-heterocystous filamentous cyanobacteria of soil crusts [34], and parallels the much more conspicuous niche differentiation of cyanobacteria known from hot springs at temperatures between 45-73 °C [67]. Similar niche separation in cyanobacterial genera are found as a function of salinity [68,69] or desiccation frequency in marine intertidal systems [70]. We could also show that the upper temperature limit for growth (and survival) under N_2_-fixing conditions is more constrained than that under non N_2_-fixing conditions (Figure 3), implicating N_2_-fixation as a possible driver of the effective upper limit of temperature range in nature. Although measurements of N_2_-fixation rates (by acetylene reduction assay or ^15^N isotopes) would give a much more direct result, the observed thermophysiological responses of the tested strains at 35 °C, coincide with more dramatic decreases in N_2_-fixation rates (above 30 °C) in cold than in hot biocrusts locations [31], and are congruent with the fact that *Nostoc* spp. and particularly *Tolypothrix* spp. are more abundant in biocrusts from colder locations, while *Scytonema* spp. typically dominate in warmer ones [32,33,34].

In an effort to better understand the basis for this effect on N_2_-fixation we determined the ratio of heterocyst frequency at different temperatures in a selected set of strains, which were responsive to our experimental conditions (*Scytonema* sp. JS006, *Nostoc* sp. HSN008 and *Tolypothrix* pp. HSN042, Figure 3). The results suggest that in *Nostoc* spp. and *Tolypothrix* spp., the impossibility of these strains to grow under N_2_-fixation conditions at temperatures above 30 °C may be determined by an inability to carry out the sophisticated developmental cycle leading to the differentiation of heterocysts [71]. While *Scytonema* spp. may have overcome such developmental problems (Table 1), nitrogenase denaturation, which has been reported to happen at temperatures above 39 °C [72] could be the basis for the observed differences in *Scytonema* spp. strains’ biomass yield at 35 °C (Figure 3). It is also possible that the observed inability of *Nostoc* spp. and *Tolypothrix* spp. strains to differentiate heterocysts at higher temperatures is the result of a resource allocation constraint to obtain the energy required to differentiate these specialized cells. However, N_2_-fixation and heterocyst differentiation at temperatures above 40 °C is not a problem in principle, in that the freshwater thermophilic cyanobacterium *Mastigocladus laminosus* performs N_2_-fixation at 45 °C [73], and is able to grow at temperatures as high as 57 °C [74]. Whether the observed heterocyst frequency decrease in *Nostoc* spp. and *Tolypothrix* spp. is a direct effect of temperature rather than a side effect due to stress on other physiological processes will need further investigation.

We tested the relevance of this temperature-based niche differentiation in nature by studying the distribution of the three cyanobacterial types as a function of climate parameters in a meta-analysis of a large dataset of biocrust surveys. Indeed, we found that the maximal relative proportion of *Scytonema* spp. among all heterocystous cyanobacteria increased along the temperature gradient with increasing temperatures (Figure 4A), when the average temperatures of the growth (wet) season was considered. Clearly, however, the results point to a potential for differential sensitivity of these cyanobacteria to environmental warming, a future scenario with which biocrust will have to contend. Drylands at large will likely become warmer and drier in response to global warming. In particular, the US Southwest is predicted to experience an increase in temperature of about 1 °C per decade [26], accompanied by alterations in precipitation frequency [35,36].

Given the observed differential response of biocrust N_2_-fixing cyanobacteria to temperature, and in agreement with Muñoz-Martín et al., (2018), it is reasonable to forecast that a microbial replacement within biocrust heterocystous cyanobacteria may indeed be in store as a result of global warming. *Scytonema* spp. may replace more cold- and mesic-temperature adapted *Tolypothrix* spp. and *Nostoc* spp. In places such as the Colorado Plateau, the Mojave desert, the north part of the Chihuahuan Desert (Sevilleta LTER) in the USA, Alicante in Spain, Western Australia [49], temperate areas in Mexico [33], and the Brazilian savannah (Cerrado) [52], where the mean annual temperature during the growth season falls between the 17 and 23 °C range, this microbial replacement will likely happen faster than at those locations exhibiting mean average temperatures below 17 °C, that are not projected to reach sensitive temperature ranges for decades to centuries, or locations with average temperatures above 24 °C, which already exhibit a dominance of *Scytonema* spp. (Figure 4). This microbial replacement could have implications for drylands and biocrust nitrogen inputs beyond a mere compositional change. *Scytonema* spp. have been shown to be one of the most sensitive taxa in biocrust to changes in precipitation patterns [47]. In this scenario, the N_2_-fixing cyanobacteria taxa that seem to be better adapted to withstand increases in temperature, ironically, seem to be among the least adapted to withstand drought. Although it makes sense that cyanobacterial distribution patterns with increasing temperature became more apparent when mean temperature during the wettest quarter of the year was used as an explanatory variable, we were surprised by the fact that plots using MAT did not show clearer patterns (Figure 4). This highlights the need to take into account the ecophysiology of microorganisms when seeking to find important climatic drivers.

These results can also serve to improve strategies to restore biological soil crust communities, of much recent interest in conservation ecology [46,75], by providing information to optimize inoculation season and microbial inoculum formulations.

## Figures and Tables

**Figure 1 microorganisms-08-00396-f001:**
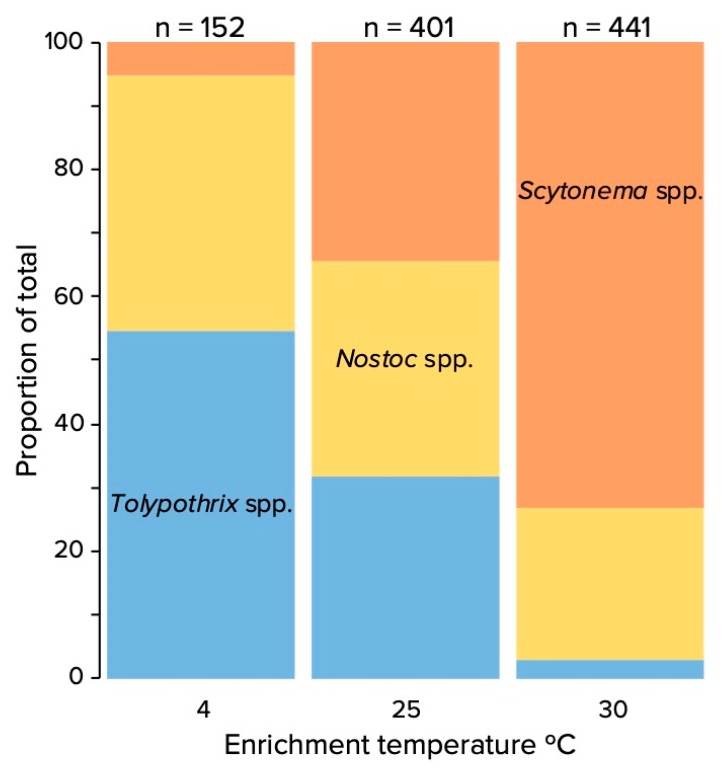
Relative proportion of colonies assignable to *Scytonema* spp., *Nostoc* spp., and *Tolypothrix* spp. in enriched cultures obtained on medium without combined nitrogen as a function of incubation temperature.

**Figure 2 microorganisms-08-00396-f002:**
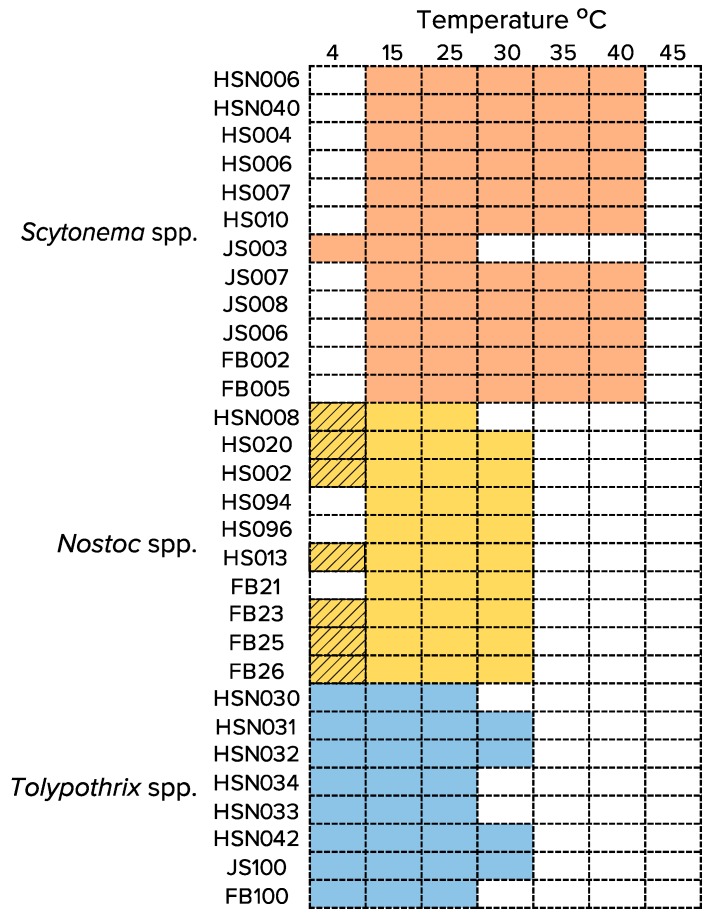
Temperature range at which the studied cyanobacterial strains can grow or survive under diazotrophic conditions. Colored rectangles indicate positive growth; hatched rectangles indicate stasis (no growth, but no obvious deterioration).

**Figure 3 microorganisms-08-00396-f003:**
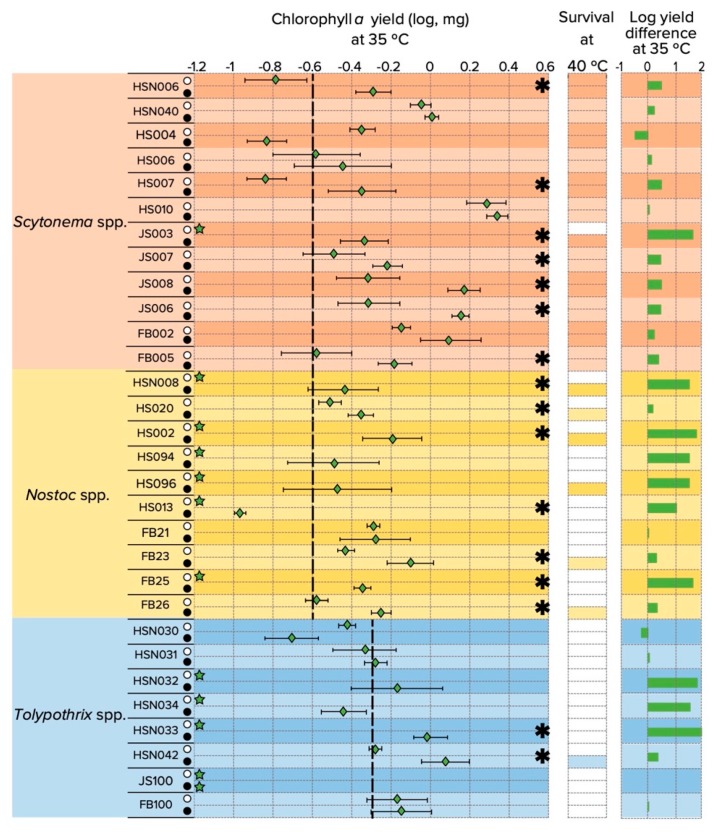
Growth yield of N_2_-fixing cyanobacterial strains in the upper range of temperature for growth in nitrogen free (⚪) vs. nitrogen replete media (●). Rhombuses indicate the mean and error bars indicate ± 1 SE, with *n* = 3. Vertical dashed lines indicate the amount of inoculum provided. At 40 °C, only observational data were recorded: colored rectangles indicate survival and white rectangles indicate death. * Denotes statistically significant differences between growing conditions according to Wilcox’s test.

**Figure 4 microorganisms-08-00396-f004:**
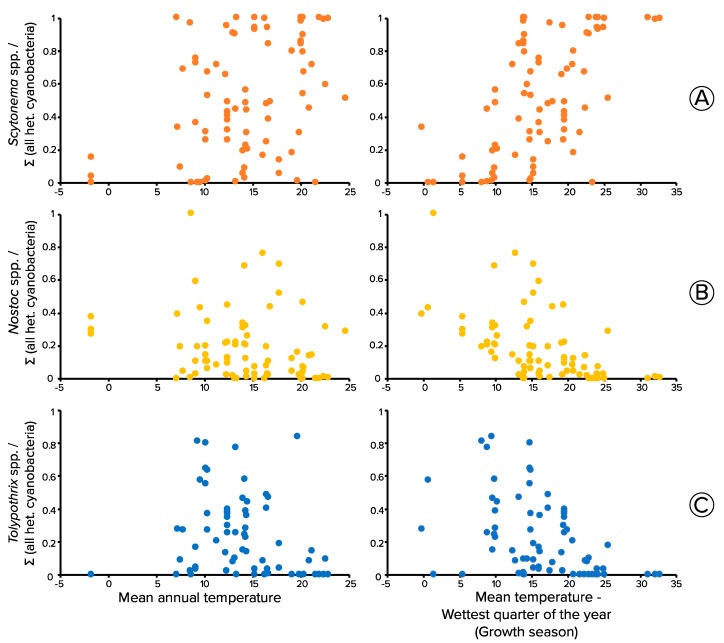
Proportion of sequence reads assignable to *Scytonema* spp. (**A**; orange), *Nostoc* spp. (**B**; yellow), and *Tolypothrix* spp. (**C**; blue) to those assignable to all heterocystous (Order Nostocales) cyanobacteria, in 16S rRNA molecular survey datasets, as a function of climate temperature indicators. Data are from biocrust communities surveyed at 84 locations around the world (see Appendix A). Each dot represents a different location.

**Table 1 microorganisms-08-00396-t001:** Frequency of heterocysts (number of vegetative cells per heterocyst) in representative cyanobacterial strains after incubation at 30, 35 and 40 °C for 7 days. Averages of *n* = 6 determinations ± standard deviation are given. H: heterocystous, VG: Vegetative cells.

Strain	Incubation Temperature °C								
	25			35			40		
	H	VG	Ratio H:VG	H	VG	Ratio H:VG	H	VG	Ratio H:VG
*Nostoc* spp.	26	230	1:9	10	207	1:21	18	569	1:32
	23	205	1:9	12	258	1:22	28	773	1:28
	32	206	1:6	11	223	1:20	21	642	1:31
	24	208	1:9	13	246	1:10	23	730	1:32
	26	207	1:8	14	236	1:17	26	701	1:27
	22	212	1:10	-	-	-	27	689	1:26
*Tolypothrix* spp.	35	467	1:13	6	628	1:105	3	900	1:300
	38	507	1:13	3	444	1:148	4	900	1:225
	40	538	1:13	2	250	1:125	2	900	1:450
	37	557	1:15	5	576	1:115	6	900	1:150
	43	613	1:14	6	553	1:92	4	900	1:225
	36	582	1:15	3	324	1:108	4	900	1:224
*Scytonema* spp.	27	534	1:20	15	352	1:23	12	526	1:44
	29	553	1:19	5	150	1:20	12	600	1:50
	32	669	1:21	7	225	1:32	10	550	1:55
	37	844	1:23	9	276	1:32	14	708	1:44
	18	308	1:17	8	242	1:30	16	641	1:50
	25	412	1:16	7	233	1:33	41	1087	1:55

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
