# Peer review of "Niche Partitioning with Temperature among Heterocystous Cyanobacteria (Scytonema spp., Nostoc spp., and Tolypothrix spp.) from Biological Soil Crusts"

_microorganisms, 2020, doi:10.3390/microorganisms8030396_

Round 1

Reviewer 1 Report

An interesting research combining laboratory experiments and analysis of molecular data obtained from natural samples of wide geography. The authors showed the ecophysiological temperature differences between three genera of heterocystous cyanobacteria Scytonema, Tolypothrix and Nostoc – the most abundant diazotrophic cyanobacteria in biological soil crusts (BSC) around the world. The obtained results allow predicting a potential changes in composition of BSCs if the global warming will effect mean temperatures during growth seasons.

Although the obtained data look pretty convincing there are several questions and suggestions to improve it.

  1. The methodology of delineation of temperature range for growth and survival of isolates is qualitative but not quantitative. It is not clear why the authors chose this method of estimation of growth while in the next experiment quantitative measurements (Chl a) were made. Although the idea of particular experiment allows using the qualitative assessments, the criteria for these evaluations should be described in more detail. What characteristics were evaluated as growth? How the stasis (no growth, but no obvious deterioration) was estimated qualitatively? How to distinguish stasis and slow growth? What was estimated as survival?
  2. The idea about the inhibitory influence of high temperature on the N2-fixation due to inability to develop heterocysts is poorly justified in the Results section. Later in the Discussian section the authors mention that the problem needs further investigation. But nevertheless the authors placed this idea to the Abstract. This is not undoubtedly. The more thoughtful statements are needed.
  3. The very interesting analysis of data from biocrust communities was made (Fig. 4), but it surely needs statistical processing to confirm the statistical significance or insignificance of trends. Also the analysis of relative proportion of sequence reads assignable to Scytonema spp., Nostoc spp., and Tolypothrix spp. at each temperature can be very useful (the same as in fig. 1). It can additionally confirm conclusions from fig. 1, but on the global scale and with the higher range of temperatures.
  4. Fig 3. Look at the figure caption: “growth in nitrogen free ( ) vs. nitrogen replete media ( )” - no icons in brackets; the meaning of asterisk is not indicated.
  5. Table 1. Abbreviations “H” and “VG” are not decrypted.
  6. Fig 4. “the total number of reads of all N2-fixing cyanobacteria” – do the authors mean only heterocystous cyanobacteria or really all N2-fixing species including unicellular and filamentous non-heterocystous taxa capable of N2 fixation?

Author Response

Reviewer #1

An interesting research combining laboratory experiments and analysis of molecular data obtained from natural samples of wide geography. The authors showed the ecophysiological temperature differences between three genera of heterocystous cyanobacteria ScytonemaTolypothrix and Nostoc – the most abundant diazotrophic cyanobacteria in biological soil crusts (BSC) around the world. The obtained results allow predicting a potential changes in composition of BSCs if the global warming will effect mean temperatures during growth seasons.

Although the obtained data look pretty convincing there are several questions and suggestions to improve it.

  1. The methodology of delineation of temperature range for growth and survival of isolates is qualitative but not quantitative. It is not clear why the authors chose this method of estimation of growth while in the next experiment quantitative measurements (Chl a) were made. Although the idea of particular experiment allows using the qualitative assessments, the criteria for these evaluations should be described in more detail. What characteristics were evaluated as growth? How the stasis (no growth, but no obvious deterioration) was estimated qualitatively? How to distinguish stasis and slow growth? What was estimated as survival?

Agreed. A more in detail explanation of the criteria used in the qualitative assessments for the temperature range for growth and survival of the studied isolates have been included in the methods sections of the manuscript. New lines 112-116.

  1. The idea about the inhibitory influence of high temperature on the N2-fixation due to inability to develop heterocysts is poorly justified in the Results section. Later in the Discussion section the authors mention that the problem needs further investigation. But nevertheless the authors placed this idea to the Abstract. This is not undoubtedly. The more thoughtful statements are needed.

Agreed. We have now toned down the reference to this phenomenon in the abstract to make it consistent with the more nuanced discussion. New lines 19-20

  1. The very interesting analysis of data from biocrust communities was made (Fig. 4), but it surely needs statistical processing to confirm the statistical significance or insignificance of trends. Also the analysis of relative proportion of sequence reads assignable to Scytonema spp., Nostoc spp., and Tolypothrix spp. at each temperature can be very useful (the same as in fig. 1). It can additionally confirm conclusions from fig. 1, but on the global scale and with the higher range of temperatures.

Agreed. Statistical analysis has been added. New information is reflected in the methods section as well as in the results section. New lines 185-187, 267-277, Tables S5-S10, Figure S1).

  1. Fig 3. Look at the figure caption: “growth in nitrogen free ( ) vs. nitrogen replete media ( )” - no icons in brackets; the meaning of asterisk is not indicated.

Missing information has been added to the figure caption.

  1. Table 1. Abbreviations “H” and “VG” are not decrypted.

Missing information has been added to the figure caption.

  1. Fig 4. “the total number of reads of all N2-fixing cyanobacteria” – do the authors mean only heterocystous cyanobacteria or really all N2-fixing species including unicellular and filamentous non-heterocystous taxa capable of N2 fixation?

We mean only heterocystous cyanobacteria, thanks for catching this misnomer. Now fixed.

Reviewer 2 Report

This is a well-written and interesting article on the segregation of ecological niches by temperature in three heterocystous cyanobacteria in BSC. The important issue is that it needs to be adapted to the current major cyanobacterial taxonomy. It is not necessary to mention in detail for classification because this is not taxonomic paper, but if the data used constitutes a monophyletic group, I recommend to indicate the representative 16S accession number for each generic group. At least in the case of Scytonema, it deals with a different phylogenetic group from the type species S. hofmanni.

Following are some minor comments.

L204, one Scytonema spp.  one Scytonema sp.

L231, Figure 3. growth in nitogen free (  ) vs. nitorogen replete media (  )

 growth in nitogen free (◯) vs. nitorogen replete media (●)

L240, Nostoc spp. HSN008  Nostoc sp. HSN008, Tolypothrix spp. HSN042 Tolypthrix sp. HSN042

L241, Nostoc spp. HSN008  Nostoc sp. HSN008

L261, no trends were conspicuous.  Should be indicated statistically.

L263, Figure 4A, Figure 4C, Figure 4B  not in the Figure 4

Figure 4. A, B, and C should be labeled. Also should be entered in the figure legend.

L295, reduction essay  reduction assay

L302, Scytonema spp. JS006 Scytonema sp. JS006, Nostoc spp. HSN008 Nostoc sp. HSN008, Tolypothrix spp. HSN042 Tolypothrix sp. HSN042

L419, ISME J. 2015, 10, 287-298, DOI: 10.1101/013813.  ISME J. 2016, 10, 287-298, DOI: 10.1038/ismej.2015.106.

L431, Biogeochemistry 2011  Biogeochemistry 2012

L441, A global approach. 2016, 161, 259-278.  A global approach. Earth-Science Reviews 161, 259–278, DOI: 10.1016/j.earscirev.2016.08.003.

L473, 2014  1984

L483-485 Should remove, the same as L480-482.

L493, 2017, 83, 1-16.  AEM. 2017, 83, 1-16, DOI: 10.1128/AEM.02179-16.

L518, DOI:10.1038/nmeth0510-335.  DOI:10.1038/nmeth.f.303.

L530-533 Should remove, the same as ref. 53.

L552-554 Should remove, the same as ref. 32.

From duplicates of references, the reference numbers should be corrected.

Author Response

Reviewer #2

This is a well-written and interesting article on the segregation of ecological niches by temperature in three heterocystous cyanobacteria in BSC. The important issue is that it needs to be adapted to the current major cyanobacterial taxonomy. It is not necessary to mention in detail for classification because this is not taxonomic paper, but if the data used constitutes a monophyletic group, I recommend to indicate the representative 16S accession number for each generic group. At least in the case of Scytonema, it deals with a different phylogenetic group from the type species S. hofmanni.

Accession numbers for the main generic groups for Scytonema spp. Nostoc spp. and Tolypothrix spp. according to our taxonomic assignment using our own cyanobacterial reference tree CYDRASIL (https://github.com/FGPLab/cydrasil) have been added as a supplementary table S4, and the manuscript has been updated accordingly. 

Following are some minor comments.

L204, one Scytonema spp. à one Scytonema sp. We have edited the text accordingly

L231, Figure 3. growth in nitogen free (  ) vs. nitorogen replete media (  )à growth in nitogen free (◯) vs. nitorogen replete media (●). Missing information has been added to the figure caption.

L240, Nostoc spp. HSN008 à Nostoc sp. HSN008, Tolypothrix spp. HSN042à Tolypthrix sp. HSN042. We have edited the text accordingly

L241, Nostoc spp. HSN008 à Nostoc sp. HSN008. We have edited the text accordingly

L261, no trends were conspicuous. à Should be indicated statistically. Agreed. Statistical analysis has been added. New information is reflected in the methods section as well as in the results section. New lines 185-187, 267-277, Tables S5-S10, Figure S1).

L263, Figure 4A, Figure 4C, Figure 4B à not in the Figure 4. Figure has been edited accordingly.

Figure 4. A, B, and C should be labeled. Also should be entered in the figure legend. Figure legend has been edited accordingly.

L295, reduction essay à reduction assay. We have edited the text accordingly

L302, Scytonema spp. JS006à Scytonema sp. JS006, Nostoc spp. HSN008à Nostoc sp. HSN008, Tolypothrix spp. HSN042à Tolypothrix sp. HSN042 We have edited the text accordingly

L419, ISME J. 2015, 10, 287-298, DOI: 10.1101/013813. à ISME J. 2016, 10, 287-298, DOI: 10.1038/ismej.2015.106. We have edited the text accordingly

L431, Biogeochemistry 2011 à Biogeochemistry 2012. We have edited the text accordingly

L441, A global approach. 2016, 161, 259-278. à A global approach. Earth-Science Reviews 161, 259–278, DOI: 10.1016/j.earscirev.2016.08.003. We have edited the text accordingly

L473, 2014 à 1984. We have edited the text accordingly

L483-485 Should remove, the same as L480-482. We have edited the text accordingly

L493, 2017, 83, 1-16. à AEM. 2017, 83, 1-16, DOI: 10.1128/AEM.02179-16.

L518, DOI:10.1038/nmeth0510-335. à DOI:10.1038/nmeth.f.303. We have edited the text accordingly

L530-533 Should remove, the same as ref. 53. We have edited the text accordingly

L552-554 Should remove, the same as ref. 32. We have edited the text accordingly

From duplicates of references, the reference numbers should be corrected. Reference numbers have been updated accordingly.